# Development and Optimization of Tamarind Gum-β-Cyclodextrin-g-Poly(Methacrylate) pH-Responsive Hydrogels for Sustained Delivery of Acyclovir

**DOI:** 10.3390/ph15121527

**Published:** 2022-12-08

**Authors:** Kanza Shafiq, Asif Mahmood, Mounir M. Salem-Bekhit, Rai Muhammad Sarfraz, Alanood S. Algarni, Ehab I. Taha, Ahd A. Mansour, Sami Al Zahrani, Yacine Benguerba

**Affiliations:** 1Faculty of Pharmacy, The University of Lahore, Lahore 54000, Pakistan; 2Department of Pharmacy, University of Chakwal, Chakwal 48800, Pakistan; 3Department of Pharmaceutics, College of Pharmacy, King Saud University, Riyadh 12234, Saudi Arabia; 4College of Pharmacy, University of Sargodha, Sargodha 40100, Pakistan; 5Department of Pharmacology and Toxicology, Faculty of Pharmacy, Pharmacy Collage, Umm Al-Qura University, Makkah 21955, Saudi Arabia; 6Medical Laboratory Science Department, Fakeeh College for Medical Sciences, Jeddah 21461, Saudi Arabia; 7Pharmaceutical Care Department, National Guard Health Affairs, Riyadh 11426, Saudi Arabia; 8Laboratoire de Biopharmacie Et Pharmacotechnie (LBPT), Ferhat Abbas Setif 1 University, Setif 19000, Algeria

**Keywords:** acyclovir, tamarind gum, β-cyclodextrin, EDX, biocompatible

## Abstract

Acyclovir has a short half-life and offers poor bioavailability. Its daily dose is 200 mg five times a day. A tamarind gum and β-cyclodextrin-based pH-responsive hydrogel network for sustained delivery of acyclovir was developed using the free-radical polymerization technique. Developed networks were characterized by FTIR, DSC, TGA, PXRD, EDX, and SEM. The effect of varying feed ratios of polymers, monomers, and crosslinker on the gel fraction, swelling, and release was also investigated. FTIR findings confirmed the compatibility of the ingredients in a new complex polymer. The thermal stability of acyclovir was increased within the newly synthesized polymer. SEM photomicrographs confirmed the porous texture of hydrogels. The gel fraction was improved (from 90.12% to 98.12%) with increased reactant concentrations. The pH of the dissolution medium and the reactant contents affected swelling dynamics and acyclovir release from the developed carrier system. Based on the R2 value, the best-fit model was zero-order kinetics with non-Fickian diffusion as a release mechanism. The biocompatibility of the developed network was confirmed through hematology, LFT, RFT, lipid profile, and histopathological examinations. No sign of pathology, necrosis, or abrasion was observed. Thus, a pH-responsive and biocompatible polymeric system was developed for sustained delivery of acyclovir to reduce the dosing frequency and improve patient compliance.

## 1. Introduction

Targeted and modulated distribution of therapeutic components is the goal of drug delivery systems [1,2]. The literature on the promotion of patient compliance includes a variety of medication administration strategies. Drug delivery systems are optimized for specific therapeutic outcomes across several biological systems to minimize the number of times a patient takes medication and to ensure that the treatment remains effective for an extended period [2].

The icosahedral capsid and double-stranded DNA make up the human herpes virus (HHV). Infection with herpes virus type 1 is common; the most recent study from the World Health Organization reported that more than 65% of the world’s population has been infected with HHV. The severity of the effects pf herpes simplex virus (HSV) on a person’s health depends on the person’s age and immune system. Herpes simplex encephalitis, gingivostomatitis, and visceral herpes infections are common in the elderly and those with weakened immune systems. HSV-1 and HSV-2 can only infect humans, as we are the ideal hosts for these viruses. Orofacial herpes infections, which manifest as blisters and sores, are most often caused by HSV-1 [3]. Virus incubation typically takes 4, but may extend as long as 12 days [4].

Acyclovir is the treatment of choice for herpes simplex virus (HSV) types I and II, as varicella zoster virus (VZV), cytomegalovirus (CMV), and Epstein–Barr virus (EBV) infections, [5,6]. It acts like guanosine and has similar properties. Acyclovir acquires a single phosphate group when thymidine kinase in the herpes virus cell wall catalyzes the reaction. The host cell’s kinase changes this monophosphate into di- and triphosphate. Triphosphate acyclovir then actively competes with and incorporates into viral DNA. Prematurely ending DNA chains blocks the replication of the genetic material [7]. For effective treatment of herpes simplex viruses 1 and 2, a typical daily dosage of 200 milligrams (mg) of acyclovir is suggested. The lowest effective dosage is classed in BCS-III (200 mg), whereas the highest effective dose is classified as BCS-IV (800 mg). The half-life is relatively short, at about 3 h. It is an ampholyte, meaning its solubility changes depending on whether the medium is acidic or basic (pKa 2.25 and pKa 9.25, respectively) [8]. Approximately 60–90% of the medication is eliminated by renal excretion. Acyclovir is poorly absorbed in oral medication, owing to its limited availability (15–30%). Therefore, high-strength acyclovir should be taken many times [9].

Polymer tamarind, a member of the Leguminosae family, is known as Imli in Pakistan. Mucilage is made up of branching carbohydrate units and originates in the endosperm. Originally from South and Southeast Asia and certain regions of Africa, this plant is now widely distributed. A thick liquid is produced when it is combined with hot water. It is used to make hydrogels, owing to their hydrophilic properties. It is pH-sensitive, has a high-quality mechanical profile, and has mucoadhesive properties. Several substances find a perfect home in it. When exposed to warm water, it forms a thick gel and is almost entirely insoluble in organic solvents [10]. This gum has been used to create many carrier systems for medication administration via the mouth, nose, eyes, and intestines. Its widespread use in drug delivery systems is marred by unfavorable properties such as rancidity, color loss, and quick breakdown in water [11].

Cyclodextrins are currently available in three distinct forms: α- (six glucose units with a tiny aperture), β- (seven glucose units), and γ- (eight glucose units with large chambers). The cavity diameter, size, and complexing nature of the β form make it popular. Complexation may alter guest molecules’ physical, chemical, and biological properties [12,13]. The inside of this cyclic conical compound is hydrophobic, whereas the outside is hydrophilic. Hydrophilic groups catalyze several chemical processes, including reduction, ester formation, polymerization, etc. [14].

Methacrylic acid (MAA) is an organic chemical that can be identified by its visual characteristics [15]: a clear, colorless liquid with a harsh and irritating odor. It dissolves in hot water and mixes with various organic solvents. Methyl methacrylate (MMA) and poly (methyl methacrylate) (PMMA) are two of MAA’s most common esters [16]. It is used extensively in hydrogels to provide them with pH-sensitive properties. Ammonium persulfate (APS) is an inorganic chemical that is a colorless, tasteless, non-irritating, and low-cost crystalline solid. It plays a key role as an initiator in various polymerization processes [17,18]. Crystalline and odorless, N, N’-methylene bis acrylamide is a white powder. A cross-linking agent forms intragel cross links by polymerizing with acrylamide to form a polyacrylamide gel. To maintain the gel’s rigidity, it might form a network instead of linear chains [19,20].

Hydrogels are very porous and can be created by a combination of physical and chemical processes; they have a strong electrical force that binds atoms to form a complex polymeric matrix. Their adaptability and intense sensitivity to their physiologic environment make them a good choice for use in medicine [21]. They can take in a considerable amount of liquid without becoming wet. The advantages of hydrogels include their excellent tensile strength, a wide range of possible uses, and their ability to absorb large amounts of water. Hydrogels have various medical applications, including in drug delivery systems, biosensors, contact lenses, tissue engineering, regenerative medicine, wound dressing, etc. [22,23]. Synthesis methods include but are not limited to free-radical polymerization, bulk polymerization, radiation cross linking, graft polymerization, etc. However, free-radical polymerization is the most common approach to generating hydrogels [24].

The aim of this study was to create a carrier system for the long-term delivery of ACV. Hydrogels were prepared by mixing TG and β-CD at various feed ratios. Swelling, sol-gel, and release were measured as a function of reaction reagent concentrations. Hematological, biochemical, hepatic, lipid, kidney, and histopathological screenings were performed as part of the toxicology study.

## 2. Results and Discussion

### 2.1. Physical Appearance

Tamarind gum, β-cyclodextrin, methacrylic acid, and methacrylic acid (MBA) were used in varying concentrations to create hydrogels. The prepared hydrogels had a pale yellow tint, a smooth texture, and a soft, elastic feel. After being dried, the hydrogels took on a pale golden hue. Each of the prepared hydrogel formulations showed exceptional durability. Figure 1 shows the final form of the generated hydrogel and the dried discs.

When the amount of MAA (3.0 to 9.0 g) was varied, the hydrogel color was transformed from milky to yellowish. However, the soft rubbery texture remained the same. When the amount of β-cyclodextrin (0.5 to 1.0 g) was increased, keeping the contents of all other ingredients constant (tamarind gum, MBA, and MAA), hydrogels with light yellow color were obtained with significant integrity. The hydrogels had a milky appearance with a rigid surface, which was achieved by increasing the amount of MBA (0.05 to 0.1 g). However, no noticeable color change was observed.

### 2.2. Fourier Transforms Infrared Spectroscopy

FTIR analysis was performed on pure β-CD, TG, MAA, MBA, ACV, and tamarind-β-CD-co-poly (methacrylate) hydrogels to determine the compatibility among ingredients and to confirm the successful grafting of polymer. FTIR scanning was performed over a range of 3500 cm^−1^ to 1000 cm^−1^. The IR spectrum of MBA was recorded by scanning in the range of 3500–1000 cm^−1^, as shown in Figure 2A. Spectral absorption bands were observed at 3305.41 cm^−1^, 1383.54 cm^−1^, 1225.23 cm^−1^, and 810.49 cm^−1^. A sharp and less intense peak was noted at 3305.41 cm^−1^, indicating the stretching vibration of the C=C bond of MBA. Other sharp peaks were observed at 1656.30 cm^−1^ and 1538.89 cm^−1^, corresponding to the stretching vibration of C=O and N-H groups, respectively.

The FTIR spectrum of the pure drug acyclovir was recorded, as shown in Figure 2B. Acyclovir presented with short and less intense peaks at 3438.11 cm^−1^ and 3179.70 cm^−1^ related to N-H and O-H stretching, respectively. C-H aliphatic stretching was observed at 2694.73 cm^−1^, a stretching vibration of the carbonyl (C=O) group was observed at 1706.71 cm^−1^, and N-H bending was observed at 1629.93 cm^−1^ [7].

The FTIR spectrum of TG is shown in Figure 2C, with prominent peaks at 1014.74 cm^−1^, 1370.89 cm^−1^, 1643.53 cm^−1^, and 3315.08 cm^−1^. A wide band was observed at 3315.08 cm^−1^, resulting from hydrogen bonding between molecules and the vibration of hydroxyl groups. Three fused peaks were observed at 1643.53 cm^−1^, corresponding to the stretching of carbonyl (C=O) groups, which can be explained by the fact that total monosaccharides of tamarind have two isomeric structures, one of which is a ring shape and the other of which possesses linear geometry. Linear structures contain variable glucose, xylose, and galactose units. These units are susceptible to carbonyl bond stretching.

The IR spectrum of pure β-CD was recorded, as shown in Figure 2D. The prominent wide band at 3292.74 cm^−1^ corresponds to vibrations of the hydroxyl (OH) group of β-CD. A short and less intense peak at 1632.66 cm^−1^ could be the result of the bending of the H-O-H group attached to its structure. The short peak at 1152.74 cm^−1^ represents the stretching of the C-H bond. A prominent sharp peak was observed at 1021.63 cm^−1^, corresponding to the stretching of C-O bonds in the ether and hydroxyl groups of β-CD.

The IR spectrum of methacrylic acid was recorded, as shown in Figure 2E, demonstrating asymmetric stretching of methyl C-H bands near wave number 2972.82 cm^−1^. In contrast, C=C stretching was observed at 1635.74 cm^−1^. A sharp peak was observed at 1708.83 cm^−1^ related to the unsaturation of the carboxylic group. A peak at 1635.74 cm^−1^ was observed in association with the stretching of the vinyl band (C=C).

To ensure polymerization, the IR spectrum of drug-unloaded tamarind-β-CD-co-poly (methacrylate) hydrogels was recorded. Results demonstrated the varying appearance of peaks compared to the parent excipients. Figure 2F shows prominent peaks at 2929.43 cm^−1^, 2322.41 cm^−1^, 1686.53 cm^−1^, 1483.01 cm^−1^, 1257.10 cm^−1^, 1169.60 cm^−1^, and 1027.09 cm^−1^. The peak at 3305.41 cm^−1^ in the IR spectrum of MBA associated with the stretching vibration of the C=C bond was shifted to a new position, i.e., 2929.43 cm^−1^. The emergence of a new sharp peak at 2322.41 cm-1 can be attributed to the stretching of the C-O bond. The peak at 1708.83 cm^−1^ resulting from the unsaturation of the vinyl group in the IR spectrum of MAA was shifted to 1686.53 cm^−1^ and widened, and its intensity increased.

Moreover, the peak at 1173.45 cm^−1^ in the IR spectrum of MAA was translocated to a new position, i.e., 1169.60 cm^−1^. The peak at 1370.89 cm^−1^ in the IR spectrum of tamarind gum resulting from symmetric and asymmetric stretching of the carbonyl (C=O) group was shifted to 1257.10 cm^−1^ in the case of the IR spectrum of the formulation. A sharp peak at 1014.74 cm^−1^ of TG resulting from the stretching vibration of the carbonyl (C=O) group was shifted to a new position, i.e., 1027.09 cm^−1^. Peak shifting, reduction in peak intensity, the vanishing of peaks, or the emergence of new peaks confirmed the formation of tamarind-β-CD-co-poly (methacrylate) hydrogels.

### 2.3. Differential Scanning Calorimetric Analysis

DSC evaluates enthalpy changes in individual ingredients, phase transition patterns, and developed formulations. DSC analysis was performed for pure ACV, β-CD, tamarind gum, and developed tamarind-β-CD-Co-poly (methacrylate) hydrogels. The DSC thermogram of acyclovir presented with a small exothermic peak at 90.88 °C and an enthalpy change of 0.003796 J/g corresponding to moisture loss, as shown in Figure 3A. A sharp and intense endothermic peak was observed at 256.80 °C, with an energy variation of 0.05210 J/g, which is related to the melting temperature of acyclovir.

The DSC thermogram of β-CD presented with an absorption of heat at 106.89 °C and an energy variation of 0.06192 J/g due to moisture loss from the polymer structure. A second peak indicating heat absorption was observed at 329.46 °C, with an energy variation of 0.003278 J/g, which corresponds to the phase transition of β-CD. An exothermic peak was observed at 382.31 °C, with an energy variation of 0.03110 J/g, highlighting the complete combustion of the polymer. The DSC thermogram of β-CD is shown in Figure 3B.

The DSC thermogram of tamarind gum presented in Figure 3C shows an exothermic event at 100.16 °C due to moisture loss from the polymer structure. An exothermic peak was observed at 325.67 °C, depicting the phase transformation of the polymer.

A DSC thermogram of tamarind-β-CD-Co-poly (methacrylate) hydrogels is presented in Figure 3D, showing an endothermic peak at 98.81 °C, followed by an exothermic peak, confirming moisture loss. A sharp and prominent peak was observed at 254.63 °C due to the melting of the polymer structure. An exothermic peak of tamarind gum completely vanished in the DSC results of the newly developed network. An exothermic peak was also observed at a high temperature of 401.38 °C, reflecting the complete combustion of the developed polymer.

### 2.4. Thermogravimetric Analysis

The study of the thermal stability of pure ingredients and fabricated products is an analytic technique used measure a sample’s weight loss (%) against increasing temperature. TGA analysis was performed on the pure drug (ACV), β-CD, tamarind gum, and the developed drug-loaded tamarind-β-CD-Co-poly (methacrylate) hydrogel network.

The TGA thermogram of ACV is illustrated in Figure 4A, with four steps of decomposition of the pure drug. Step I showed initial decay due to the escape of water, with a weight loss of 1.09% at 99.01 °C. Step II revealed a mass loss of 5.45% at 172.48 °C. Step III revealed a mass loss of 6.11% at 268.69 °C. Step IV showed the complete combustion of drug corresponding to a mass loss of 10.79% at a temperature of 305.65 °C.

The TGA thermogram of TG exhibits the percentage weight loss in four stages, as shown in Figure 4B. Step I indicated a weight loss of 10.45% at 110.31 °C due to moisture loss. Step II revealed a mass loss of about 19.6% at 299.54 °C due to the breakdown of weak glycosidic bonds. Step III revealed a mass loss of 61.27% at 360.45 °C. Step IV indicated the complete combustion of polymer, with a mass loss of 79.95% at a temperature near 501.10 °C.

The TGA thermogram of the β-CD exhibits a mass loss event in three phases, as shown in Figure 4C. Phase I showed an initial weight loss of 13.03% at 109.44 °C, indicating moisture content. Furthermore, phase II showed weight loss of 18.35% at 327.46 °C, which might be due to the polymer’s breakdown of hydrogen bonding. Phase III indicated total decomposition of the polymer, with a weight loss of 84.28% at 359.21 °C.

The TGA thermogram of the drug-loaded developed network was recorded, and results are shown in Figure 4D. More than 60% mass remained intact, even at temperatures above 300 °C. A further temperature increase led to the decomposition of the developed network. Even at elevated temperatures, i.e., above 350 °C, nearly 20% of the mass was intact, indicating the stability of the developed network relative to that of the individual feed ingredients.

### 2.5. XRD Diffraction Analysis

XRD studies were performed to verify the nature of the ingredients, including whether they are amorphous or crystalline. Crystalline samples presented with poor solubility and dissolution profiles, offering poor bioavailability. In contrast, samples with an amorphous nature exhibited more solubility and good dissolution profiles. TG, β-CD, ACV, and tamarind-β-CD-Co-poly (methacrylate) hydrogels were scanned in the range of 0–80°.

The XRD diffractogram of ACV is shown in Figure 5A. Prominent and intense peaks were observed at angles of 2θ = 23.45°, 26.63°, 26.29°, and 31.45°. A few less prominent peaks were also observed at 37.95°, followed by the last peak at 47.71°.

The XRD diffractogram of tamarind gum was also recorded at a scanning range of 0–70°, with an intense/sharp peak at an angle of 2θ = 19.54°, as shown in Figure 5B. The peak indicates that tamarind gum retained its structural integrity, confirming the partial crystalline nature of the substance.

Similar results were reported in a study by Kumar et al. (2018) in which a tamarind gum biopolymer membrane was prepared. XRD findings revealed a broad, intense peak of 19.97° at 2θ, confirming the semicrystalline nature of tamarind gum [25].

The XRD diffractogram of β-CD was also recorded at a scanning range of 20–90°, as shown in Figure 5C. The crystalline nature of β-CD was evidenced by the peaks at angles of 2θ = 20.65°, 27.71°, 25.91°, 41.23°, and 45.21°. Similar peaks were reported in the literature regarding β-CD. Therefore, the existence of these peaks confirms the crystalline nature of β-CD.

The XRD diffractogram of the developed formulation (TGB12) was also recorded, with peaks of β-CD at angles of 2θ = 20.65°, 27.71°, 25.91°, 41.23°, and 45.21° and peaks of acyclovir at angles of 2θ = 23.45, 26.63, 26.29, 31.45°, 37.95°, and 47.71°, which were not present in the diffractogram of drug-loaded developed hydrogel networks, as shown in Figure 5D. This indicates that the formulations were transformed from a crystalline nature to an amorphous nature. The drug incorporated into the developed network was converted into an amorphous state with an excellent dissolution profile.

### 2.6. Energy-Dispersive X-ray Spectroscopy

EDX is a technique used to confirms the elemental composition of test substances, providing the overall composition of the sample. The EDX spectra of acyclovir and unloaded and acyclovir-loaded networks were determined. Quantitative data confirm the presence of carbon, nitrogen, oxygen, and potassium in all tested samples. The elemental composition of ACV, as well as drug-loaded and unloaded formulations, is shown in Table 1.

The EDX spectrum of acyclovir revealed the presence of carbon, nitrogen, and oxygen in variable concentrations, i.e., 39.85%, 30.98%, and 28.25%, respectively. These elements were also a part of the chemical structure of acyclovir. When the EDX spectrum of the unloaded developed network was recorded, it revealed only the presence of carbon and oxygen in varying concentrations of 58.81% and 41.19%, respectively. These elements were also present in the chemical formula of reactants involved in hydrogel synthesis. In the EDX spectrum of acyclovir-loaded hydrogels, carbon (61.16%), nitrogen (10.39%), oxygen (38.45%), aluminum (0.11%), silicon (0.14%), and magnesium (0.08%) were found. Moreover, a peak of nitrogen, an essential part of acyclovir, was not observed in the unloaded hydrogel, although a nitrogen peak was observed in the case of acyclovir-loaded hydrogels, confirming the successful loading of acyclovir in the prepared copolymeric networks. Elemental presentation according to ingredient type is shown in Figure 6, with respective concentrations shown in Table 1. Mahmood et al. (2018) prepared acyclovir-loaded polymeric networks and evaluated their developed formulations by EDX. The results revealed the presence of nitrogen element in the acyclovir-loaded formulation, as observed in our study [26].

### 2.7. Scanning Electron Microscopy

SEM was used to study the surface morphology and structure of the developed formulation at varying magnification powers, i.e., 100×, 250×, 500×, 1000×, 2500×, and 5000×, to confirm the network topology. All surfaces of the developed tamarind gum were smooth, slightly wavy, and porous, as shown in Figure 7. Porous surfaces facilitate processes such as the uptake of media in terms of swelling, drug loading, and release of active therapeutic agents in solution form.

### 2.8. Determination of the Sol-Gel Fraction

The sol-gel fraction was determined was performed in all developed formulations (TGB1–TGB12) to determine the cross-linked and uncross-linked fractions of the ingredients. The gel fraction (%) confirms that many reactants were consumed in the developed tamarind-β-CD-co-poly (methacrylate) hydrogels. The sol fraction (%) indicates that non-reactive soluble species did not participate in polymerization reactions. The impact of varying concentrations of components on gel fractions (%) was assessed, as graphically presented in Figure 8.

Tamarind-gum-containing formulations (TGB1–TGB3) with variable quantities (0.2 to 0.4 g) exhibited an increased gel fraction, i.e., from 90.12 to 97.12%. Tamarind gum is not a gelling agent. At high temperatures, its galactopyranosyl residues are removed, which enhances its adhesiveness and imparts a gelatinous character at higher temperatures, increasing the viscosity of the polymerization solution. Moreover, it contains numerous hydroxyl (OH) groups, which are prone to covalent bonding, resulting in increased cross linking and hence cross-linking density of networks [27].

β-Cyclodextrin containing formulations (TGB4–TGB6) with variable quantities (0.5 to 1.0 g) presented with an increased gel fraction, i.e., from 91.33 to 97.88%. When β-CD contents were increased, a saturated solution formed, owing to the poor solubility of the β-CD solution. Moreover, many hydroxyl (-OH) groups were present for cross linking, promoting the gel fraction.

MBA formulations (TGB7–TGB9) with variable quantities (0.05 to 0.1 g) resulted in an increased gel fraction, i.e., from 91.82 to 98.12, as a result of the increased MBA concentrations caused by increased interactions between polymeric and monomeric units in the form of cross linking. By increasing MBA content, cross linking was promoted, owing to the reduced pore size of the developed networks. As a result, hydrogels with a compact and hard structure were obtained. The increased MBA content also resulted in an enhanced cross-linking density of the network.

MAA formulations (TGB10–TGB12) with variable quantities (3.0 to 9.0 g) exhibited an increased gel fraction, i.e., from 90.28 to 95.66%. When the amount of MAA was increased, the availability of carboxylate ions was increased for cross linking. As a result, the network’s cross-linking density and integrity were increased.

### 2.9. Equilibrium Swelling Studies

Swelling studies were conducted to confirm the stimulus-responsiveness of the developed network, which is an integral parameter related to drug loading and release of therapeutic agents from the entangled polymer structure. All the developed formulations (TGB1–TGB12) were subjected to swelling equilibrium studies at pH 1.2 and 7.4. The impact of varying quantities of ingredients, i.e., TG, β-CD, MBA, and MAA, on swelling was carefully monitored. All formulations exhibited considerable swelling at high pH values; on the other hand, only slight or minor swelling, i.e., less than 5%, was observed at low pH values, owing to the lack of ionization of pendant functional groups from polymeric units. The response of the developed network against acidic and basic pH is presented in Figure 9.

Formulations (TGB1–TGB3) containing increasing contents of tamarind gum exhibited decreased swelling at pH 7.4, i.e., from 83.69 to 73.31%, which may be related to increasing tamarind concentrations (0.2 to 0.4 g) and increased viscosity of the polymer solution. Pronounced polymeric entanglement promoted cross-linking network density. As a result, a denser network was formed [27].

Evaluation of swelling behavior of β-CD-containing formulations (TGB4–TGB6) with variable quantities (0.5 to1.0 g) revealed an increasing trend of equilibrium swelling, i.e., from 61.51 to 79.57% at pH 7.4 and hydroxyl (-OH) groups β-CD with strong hydrophilic character. The availability of additional ionized hydroxyl (-OH) groups contributed to optimum swelling resulting from the repulsive forces generated between hydroxyl (-OH) and carboxylate (COO-) ions. Uyanga et al. (2020) also highlighted that numerous hydroxyl groups in the β-CD structure promote linking and the formation of a stable network [28].

Evaluation of the swelling behavior of MBA-containing formulations (TGB7–TGB9) with variable quantities revealed a decrease in equilibrium swelling, i.e., from 86 to 67.23% at pH 7.4 when the amount of MBA was increased from 0.05 to 0.1 g. This decreased swelling may be the result of increased cross linking, resulting in increased cross-linking density. As a result, a denser, more compact, and hard structure was obtained. Owing to the dense structure, pore size was markedly reduced. As a result, the uptake of swelling media or biological media was reduced, resulting in limited swelling.

Evaluation of swelling behavior of MAA-containing formulations (TGB10–TGB12) with variable quantities revealed an increasing trend in equilibrium swelling, i.e., from 79.54 to 91.61% at pH 7.4 when the amount of MAA was increased from 3.0 to 9.0 g. At basic pH, protonated carboxylic groups of MAA were converted to deprotonated chains, resulting in the generation of an electrostatic force of repulsion between polymeric chains, creating spaces between chains, which are helpful for the uptake of swelling media. Results presenting the effect of ingredients on swelling studies are shown in Figure 10. Garcia et al. (2004) reported similar trends in swelling in response to increasing MAA contents [29].

### 2.10. Effect of Feed Content on Acyclovir Release

The contents of ACV were released at pH 1.2 and 7.4 from all of the produced formulations (TGB1–TGB12). At pH 1.2 and 7.4, the behavior of ACV-loaded discs was investigated to determine the impact of TG, β-CD, MBA, and MAA on drug release. USP type-II dissolution (paddle) equipment was used in a 37 °C temperature-controlled dissolution experiment. All formulations showed some or no drug-release behavior at a pH of 1.2 (TGB1–TGB12). Drug release ranged from 73.12 to 93.44% across all formulations at pH 7.4. Formulation TGB12, which had the highest concentration of MAA, showed the highest rate of release (93.44%), which may be attributable to the ionization of the carboxylic acid in MAA.

When TG content was increased from 0.2 to 0.4 g per mL, the proportion of ACV release was reduced from 81.55 to 75.5%. (TGB1–TGB3). Figure 11 displays the obtained outcomes. Possible causes include the water-repellent properties of the hydrogel, as well as its dense network, which reduces swelling. Our results corroborate those reported by Mali (2018), who found that increasing the tamarind gum dosage reduced the drug’s bioavailability [30].

With an increase from half a gram to a gram of β-CD, the percentage of drug release increased from 78.64 to 89.87% (TGB4–TGB6). Figure 11 displays the obtained outcomes. Because β-CD contains more ionizable hydroxyl groups and is hydrophilic, the ionization of these hydroxyl groups causes the polymeric chains to resist one another, making room for the absorption of swelling media. Similar to the results reported in this investigation, Malik et al. (2017) created polymeric networks in which drug release improved when the quantity of β-CD was increased [6].

At a pH of 7.4, increasing the MBA concentration from 0.05 to 0.1 g decreased drug release from 83.72 to 73.12% (TGB7–TGB9). Figure 11 displays the obtained outcomes. The high binding capacity may result in a thick, robust hydrogel with entangled molecules, preventing drug release from the constructed network.

Hydrogels containing galantamine hydrobromide were created by Bashir et al. (2020). According to their results, increasing the concentration of MBA resulted in fewer drugs being released. Our findings mirrored those of the previous research [31].

An increase in MAA concentration from 0.0 to 9.0 g (TGB10–TGB12) enhanced drug release from 87.45 to 95.0% at a pH of 7.4. Figure 11 displays the obtained outcomes. One possible explanation is that the carboxyl group of MAA is ionized at high pH levels, causing the molecules to unwind and allowing for additional drug release.

Khan et al. (2017) created a polymeric carrier system to effectively load the medication metoprolol, which dissolves in water. The effect of the system on medication release was investigated. We previously [32] found that a increase in MAA concentration led to a more rapid drug release.

### 2.11. Drug Release Kinetics

Kinetic models, including the zero-order, first-order, Higuchi, and Korsmeyer–Peppas models, were applied to the release data of all formulations (TGB1–TGB12) to determine the best-fit model based on the R^2^ value and the mechanism of ACV release from the produced formulations. All formulations (TGB1–TGB12) were found to follow zero-order kinetics, as predicted by the results (Table 2). This demonstrates that ACV is effective against viral infections when administered continuously. The method of ACV release was confirmed by analyzing the exponent “*n*” value. According to the mean calculated value of “*n*”, all the formulations involve Super Case II as an ACV release mechanism. A non-Fickian diffusion mechanism for ACV release was involved. All formulations except TGB1, TGB11, and TGB12 used Super Case II transport as the ACV release mechanism, as shown by the fact that “*n*” was larger than 0.89. In contrast, “*n*” values for formulations TGB1, TGB11, and TGB12 ranged between 0.5 and 0.89 according to Case II transport.

### 2.12. Toxicity Studies

Research on the acute oral toxicity of tamarind-β—CD-co-poly (methacrylate) hydrogels were conducted in albino rabbits according to the standards set out by the Organization for Economic Co-operation and Development (OECD). The Institutional Research Ethics Committee of the School of Pharmacy, University of Lahore, examined and authorized all studies before they were conducted (notification number: IREC-2021-10). The rabbits were all in good condition and ranged in weight from 1500 to 1900 g. Group I was the control, and group II comprised the experimental subjects (Treated). For 14 days, we kept careful notes on everything that happened.

#### 2.12.1. Clinical Manifestations

Every day, the rabbits were examined by hand for any visible symptoms of illness or abnormality. For 14 days, researchers tracked vitals such as heart rate, breathing rate, body mass index, skin or eye toxicity, mortality, and dietary and fluid consumption. No individuals in either group seemed to change physically throughout the study period. Table 3 displays the obtained data.

#### 2.12.2. Blood Analysis

Blood samples from both groups (control and treated) were withdrawn for hematological and biochemical screening on the 1st day and 14th day. Results are displayed in Table 4. Hematological and biochemical findings in both the control and treated groups fall within standard reference ranges. There was no change in the blood profile of the treated group after feeding with the developed polymeric network. On a hematological and biochemical level, the results show that tamarind/β—CD-co-poly (methacrylate) hydrogels are safe for use. Table 5 displays the results of the LFT, RFT, and lipid profiles in both the control and treatment groups. The liver profile was found to have undergone some minor changes. However, results were found to be within normal ranges. The renal and lipid profiles of both groups were observed to be within reference ranges.

#### 2.12.3. Histopathological Evaluation

On the 14th day, rabbits from both groups were sacrificed, and their vital organs were removed for histopathological assessment. Histology slides of each vital organ were prepared. Histopathological examination results are shown in Figure 12. No changes were observed between the control and treated groups. Figure 12A,G indicate normal brain cells of the control and treated groups with clearly visible neuronal cell bodies. No sign of immune infiltration or inflammation was observed. Brain tissue was normal with axonal cells and oligodendrocytes. Figure 12B,H show lung tissues of the control and treated groups. Normal alveoli were observed. No sign of inflammation or cellular damage was observed. Figure 12C,I show the cardiac tissue of both the control and treated groups. No inflamed or necrotic areas were observed. Cardiac cells were normal in size. Figure 12D,J indicate the liver tissue section of the control and treated groups. Normal hepatic cells were observed. No sign of inflammation or degradation of cells was observed.

Figure 12E,K show the kidney tissues of the control and treated groups. Normal cellular morphology of the glomerulus and Bowman’s capsule was noted. No sign of localized or general inflammation in the tubular or ductal cells was observed. Overall, cells appeared normal in shape. Figure 12F,L show a section of intestine of both the control and treated groups. The circular smooth muscle layer and the serosal layer were intact. Normal morphology of epithelial cells was observed. Hyperplasia was not observed. Columnar epithelial cells appeared normal.

Asghar et al. (2021) developed spongy matrices using the free-radical polymerization method and a formulation using a polymer called β-cyclodextrin. According to OECD guidelines, they conducted an acute oral toxicity study on rabbits. No evidence of inflammation, abnormalities, or alterations in anatomy or physiology was found upon histopathological investigation, indicating that these sponge-like matrices are safe for use [33]. Acute oral toxicity tests of the constructed networks yielded similar results.

## 3. Materials and Methods

Acyclovir was received as a generous gift from Brooks Pharmaceuticals, Pakistan. Tamarind seed powder was procured from BDH labs, England. Methacrylic acid, ammonium persulfate, β-cyclodextrin, and N, N-methylene bis-acrylamide were acquired from a US-based company called Sigma-Aldrich. All ethanol, potassium dihydrogen phosphate, and sodium hydroxide were obtained from the German company Merck. Silica gel was obtained from a local company called Naseer Sons. Distilled water was freshly prepared in the Postgraduate Research Lab of the Faculty of Pharmacy, University of Lahore.

### 3.1. Methods

#### 3.1.1. Method of Extraction of Tamarind Mucilage

Crushed tamarind seeds (about 20 g) were soaked in 200 mL of cold distilled water for 1–2 days. Approximately 800 mL of distilled water was poured into a 1000 mL beaker and boiled in a water bath (Memmert, made in Germany). Powder-containing slurry was transferred into boiling water and stirred continuously with a glass rod until a viscous solution was obtained. The resultant solution was kept overnight. The next day, the solution was centrifuged at 5000 rpm for 10–20 min. To produce tamarind gum polymer, the supernatant was collected and treated with double the amount of pure ethanol (95%). Later, it was filtered with a duly folded muslin cloth. The residue was poured into dry and clean Petri dishes and subjected to drying at 40–50 °C for 2–3 days. After complete drying, it was ground with a pestle and mortar. Gum powder was preserved in an airtight container at room temperature [34,35].

#### 3.1.2. Development of Tamarind-β-CD-co-Poly (Methacrylate) Hydrogels

The aqueous free-radical polymerization method (Figure 13) was employed with slight modifications in terms of temperature, mixing sequence, and drug loading to prepare tamarind-gum-based polymeric networks. The required amount of each ingredient consumed in all formulations, i.e., TGB1–TGB12 (Table 6), was precisely weighed on an electronic weighing balance (DV215CD, OHAUS, Cooperation USA). The weighed amount of tamarind gum powder was transferred into centrifuge tubes containing water (7–10 mL) and centrifuged at 5000 rpm for 4–5 min. The supernatant was collected using a glass pipette and transferred into a beaker (A). Acyclovir (ACV) powder was dissolved in a water–ethanol mixture in another beaker (B) and covered with aluminum foil and stirred with a magnetic hot plate stirrer (J-HSD180, JISICO Korea) at 50 °C and 100 rpm until the formation of a clear solution. While stirring, a few drops of 0.2 M NaOH were added to facilitate the complete solubility of ACV. A weighted amount of β-cyclodextrin was added to the ACV ethanol solution under stirring. Pure tamarind gum solution from beaker A was transferred to the above solution, along with ammonium persulfate as an initiator, to generate active sites. In another beaker, MBA was predissolved in water and then in methacrylic acid, and this solution was added dropwise to the solution mentioned above.

Stirring was continued until a clear solution was achieved, which was then vortexed and transferred to clean and dry labeled glass test tubes. Test tubes were sealed with aluminum foil and kept in a water bath at 60–65 °C for 2–3 h. Test tubes were taken out upon hardening to cool down to room temperature. With the help of a microspatula, hydrogel rods were carefully removed and transferred to washed and labeled Petri dishes. Prepared hydrogel rods were cut into 1-inch discs with the help of a sharp-edged blade. Discs were washed with water and methanol solution (70:30) to remove unreacted species. The pH of the solution was checked periodically until a constant pH value was not attained. After complete washing with the methanol–water mixture, discs were removed, placed into Petri dishes, and subjected to drying in a hot air oven (Memmert) for 2 days at 40 °C. Dried hydrogel discs were kept in airtight containers for further analysis [36]. The proposed chemical structure of the developed hydrogels is shown in Figure 14.

### 3.2. Characterization

#### 3.2.1. Fourier Transform Infrared Spectroscopy

FTIR spectra of β-CD, tamarind gum powder, MAA, ACV, and the developed hydrogels were recorded to confirm complex formation and verify the ingredients’ compatibility. The attenuated total reflectance (ATR) technique was employed. A Bruker FTIR spectrophotometer (Tensor 27 series; Bruker Co. 182 Germany) was used with OPUS data collection software. Samples were ground to a powder, placed on the lens, and scanned in the range of 3500 cm^−^^1^ to 1000 cm^−^^1^ [37].

#### 3.2.2. Differential Scanning Calorimetry

DSC thermal analysis was carried out to investigate the thermal transitions of β-CD, tamarind gum powder (TG), ACV, and drug-loaded formulations. Samples were pulverized and covered in an aluminum pan. To this end, SDT (Q600 TA USA) was used at 10 °C/min under a nitrogen atmosphere in the temperature range of 0–400 °C. The heating rate was maintained at 20 °C/min. Every sample was scanned three times, and DSC thermograms were recorded [38,39].

#### 3.2.3. Thermogravimetric Analysis

TGA studies were executed to check the thermal stability of the formulated network compared to that of each ingredient consumed in the development of the hydrogel, e.g., β-CD, TG, MAA, MBA, and the formulation. Samples were triturated and analyzed by SDT (Q600 TA USA) at a scanning rate of 20 °C/min under previously described conditions. Changes in mass with increasing temperature were recorded [38,40].

#### 3.2.4. Scanning Electron Microscopy

The surface morphology of the formulated network was investigated using an electron microscope (JSM-5910 instrument, JEOL, Japan). Dried discs were ground to mount on a double-adhesive tape on an aluminum stub. Photomicrographs were captured at varied magnifications, i.e., 10,000×, 5000×, 4000×, 2000×, etc. [37,41].

#### 3.2.5. X-ray Diffraction Studies

The nature of the ingredients, whether crystalline or non-crystalline/amorphous, can be confirmed by X-ray diffraction. The developed network, ACV tamarind gum powder, and β-CD were subjected to scanning by an X-ray diffractometer (JDX-3532 JEOL Japan) with a CuKα radiation source. A voltage of 40 kV and a 30 mA current were used. The scanning range was set to the range of 10° to 70°. Every sample was scanned three times, and peaks were recorded [42].

#### 3.2.6. Energy-Dispersive X-ray Spectroscopy

EDX (Inca 200 m Oxford, UK) is an analytical technique that evaluates the elemental composition of ingredients and developed networks and confirms the successful loading of drugs within the network. EDX spectra of acyclovir-unloaded hydrogel and the acyclovir-loaded network were investigated and compared [43].

#### 3.2.7. Sol-Gel Fraction

The sol-gel fraction of the polymeric network was determined out in boiling water using a Soxhlet apparatus. This process removes reactants that were not cross-linked during the network synthesis. The gel fraction indicates the amount of reactants consumed in the preparation of tamarind/β-CD-co-poly (methacrylate) hydrogels. Sol contents are the unreacted soluble contents of free-radical polymerization reactions. Dried hydrogel discs were weighed, crushed into small pieces, and placed in boiling water using a Soxhlet apparatus. After 4 h, pieces were removed on filter paper, dried in an oven at 40–45 °C, and weighed again [41,44]. The sol-gel fraction was calculated by applying the following equations:$$Sol\;fraction\;\left(\%\right)=\frac{Wo-Wt}{Wo}\times 100$$
where *Wo* is the initial mass of the dried discs before extraction, and *Wt* is the final mass after extraction.
$$Gel\;fraction\;\left(\%\right)=100-sol\;fraction$$

#### 3.2.8. Swelling Studies

To evaluate the pH-responsive behavior of tamarind/β-CD-co-poly(methacrylate) hydrogel, its swelling behavior was studied at pH 1.2 and 7.4. Dried discs were weighed and poured in a freshly prepared phosphate buffer solution with a pH of 7.4 and an acidic buffer solution with a pH of 1.2 and kept separately at 37 °C. Discs were taken out of the solution at regular intervals (i.e., 0.5, 1, 1.5, 2, 3, 4, 5, 6, 12, 18, 24, 30, 36, 42, and 48 h) blotted with filter paper to remove extra liquid from surface, and weighed accurately.

The following formula was used to calculate equilibrium swelling (*ES*):$$ES=\frac{W_{t-}W_{i}}{W_{i}}\times 100$$
where *ES* = equilibrium swelling, *Wi* = initial weight of the hydrogel, and *Wt* = weight of the swollen hydrogel [45].

#### 3.2.9. In Vitro Drug Release Studies

First, 3, 5, 7, 9, 11, 13, and 15 μg/mL ACV dilutions were prepared from the stock solution (500 µg/mL) of the drug. After appropriate dilution, samples were scanned on a UV-Vis spectrophotometer to determine the absorbance results. Scanning mode was used to find the λ_max_ value, which was found to be 251 nm. The absorbance of all prepared dilutions was recorded on scanned λ_max_, i.e., 251 nm. The calibration curve was constructed using concentration values against corresponding absorbance values. A straight-line equation and regression coefficient (R2) of 0.9959 was recorded.

The in vitro drug release behavior of acyclovir-loaded hydrogels was investigated on a USP dissolution apparatus type-II with a paddle speed of 50 rpm at 37 °C. Phosphate buffers of with pH of 1.2 and 7.4 were employed as dissolution media for release experiments.

Drug-loaded hydrogels were placed in baskets containing 900 mL of dissolution media. At fixed time intervals (0, 0.5, 1, 2, 3, 4, 6, 8, 10, 12, 15, 18, 21, and 24 h), 5 mL of the sample was collected using a glass pipette from the center of the vessels, and 5 mL of fresh buffer was poured in to maintain sink conditions. Samples were then filtered and scanned to obtain absorbance results at each time interval. Unknown concentrations in µg/mL were calculated by a straight-line equation, and ACV release was recorded with respect to time [46,47].

#### 3.2.10. Kinetic Modeling

Acyclovir release data from tamarind-β-CD-Co-poly (methacrylate) hydrogels were processed through kinetic models, i.e., zero-order, first-order, Higuchi, and Korsmeyer–Peppas models, with the help of the DD solver^®^ MS Excel add-inn program. The R^2^ value was determined to identify the best-fit model and mechanism of acyclovir release according to drug release exponent (*n*). The equations of each model are given below:

Zero-order kinetics
$$Mt=K_{o}t$$

First-order kinetics
$$M_{t}=1-e^{-K_{1}t}$$

Higuchi model
$$Mt=KH^{1/2}$$
where *Mt* = the fraction of released drug at each time point (*t*), *K_o_* = the zero-order rate constant, *K*1 = the first-order rate constant, and K_H_ = the Higuchi release kinetic constant.

The drug release exponent (*n*) was calculated using the Korsmeyer–Peppas model, the equation of which given below:$$\frac{M_{t}}{M_{\infty }}=Kt^{n}$$
where *Mt* = amount of absorbed water at any time (*t*), *M*∞ = mass or water uptake at equilibrium, *n* = exponent explaining the swelling mechanism, and *K* = rate constant. If *n* = 0.45, it follows Fickian diffusion; n values ranging between 0.45 and 0.89 indicate non-Fickian or anomalous diffusion. An n value greater than 0.89 indicates Super Case Type-II release mechanism [48].

#### 3.2.11. Oral Acute Toxicity Studies

Economic Co-operation and Development (OECD) guidelines for testing chemicals were strictly followed for conduct toxicity studies. Twelve healthy albino rabbits of average weight were purchased from the Animal House of the Faculty of Pharmacy, University of Lahore. Study protocols were reviewed and approved by the Institutional Research Ethics Committee of the Faculty of Pharmacy, University of Lahore (notification no. IREC-2021-10). Before initiating the study, rabbits were kept in clean wooden cages for one week to acclimatize them to the environmental conditions in a 12 h light and dark cycle. They were randomly divided into groups I and II (n = 6) and were provided feed per standard laboratory diet plans, with free access to water. The treated group was administered an optimized formulation (TGB12) at a dose of 2 g/kg. The study was continued for 14 days.

##### Sampling

Rabbits were fasted for approximately 12 h before administration of the carrier system, with free access to water to reduce any possible chances of drug–food interaction. The weight of each rabbit was recorded. Blood screening was performed on all rabbits, including CBC, LFT, RFT, and lipid profile. Group I was labeled as the control, and group II was administered tamarind-β-CD-co-poly (methacrylate) hydrogel in crushed form.

##### Clinical Manifestations

All rabbits were monitored daily to check for any signs of sickness/ailment, e.g., stomach upset, watery mouth, eye irritation, sleep changes, skin rash, behavioral changes, convulsions, loss of appetite, etc.

##### Blood Analysis

Blood samples were taken from the supraorbital vein of all rabbits on the 1st day and 14th day for biochemical and hematological screening. Blood samples were collected in ethylene diamine tetra acetic acid (EDTA)-K2 blood collection tubes. A hematological analyzer (ABX Pentra 60; Horiba, Japan) was used to assess the blood samples for complete blood count (CBC). The rest of the blood samples were centrifuged to separate serum for biochemical analysis using a chemical analyzer (Erba Mannheim XL- 200; Brno, Czech Republic).

##### Histopathological Examination

On the 14th day, animals were sacrificed, and vital organs such as the heart, brain, liver, kidney, spleen, lungs, intestine, and stomach were removed, weighed, and kept in labeled formalin (10%)-containing jars after washing. Tissues were fixed on paraffin blocks. Tissue segments were removed from these blocks and fixed on glass slides. Histology slides were subjected to hematoxylin and eosin (H&E) staining. An optical light microscope (Nikon E200, Tokyo, Japan) was utilized to capture images with a built-in camera (DCM-35 USB 2.0 and MINISEE IMAGE software, Scopetek Electric, Hangzhou, China) [31].

## 4. Conclusions

A novel tamarind gum-β-CD-g-poly(methacrylate) hydrogel was successfully developed through free-radical polymerization and optimized for controlled delivery of acyclovir. We anticipate that tamarind-β-CD-Co-poly(methacrylate) hydrogels hold considerable potential for bioavailability enhancement and sustained drug delivery of the anti-viral agent acyclovir, reducing the required dosing frequency and ultimately enhancing patient compliance.

### 4.1. Limitations of the Study

Drug loading of hydrophobic drugs is difficult. Carrier system washing to remove unreacted species is a time-consuming process, and it is difficult to load low doses of drugs. The results of the present study cannot confirm the therapeutic efficiency of acyclovir.

### 4.2. Future Prospects

This study can be extended to the pharmacokinetic evaluation of acyclovir in a suitable animal model using a validated HPLC method. Moreover, herpes can be induced in an animal model to verify the therapeutic efficiency of acyclovir.

## Figures and Tables

**Figure 1 pharmaceuticals-15-01527-f001:**
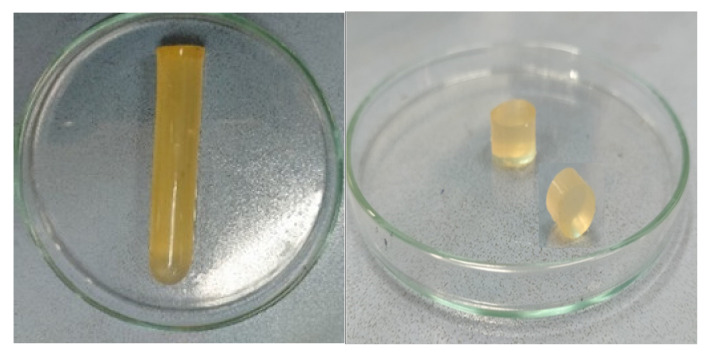
Physical appearance of hydrogels.

**Figure 2 pharmaceuticals-15-01527-f002:**
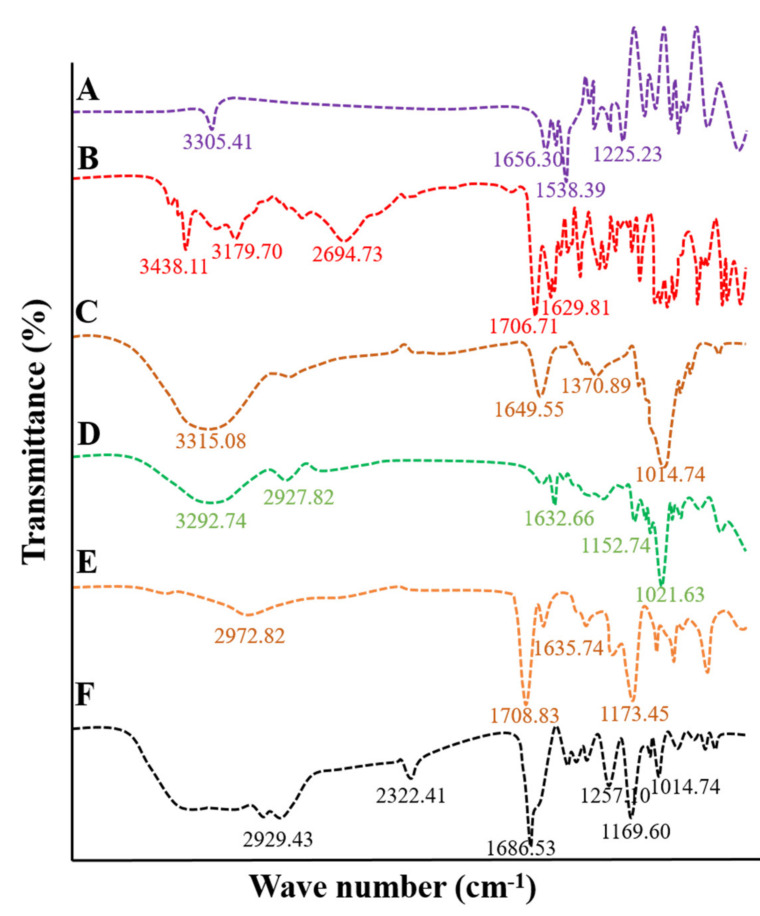
FTIR spectra of (**A**) MBA, (**B**) acyclovir, (**C**) tamarind gum, (**D**) β-CD, (**E**) MAA, and (**F**) formulation TGB12.

**Figure 3 pharmaceuticals-15-01527-f003:**
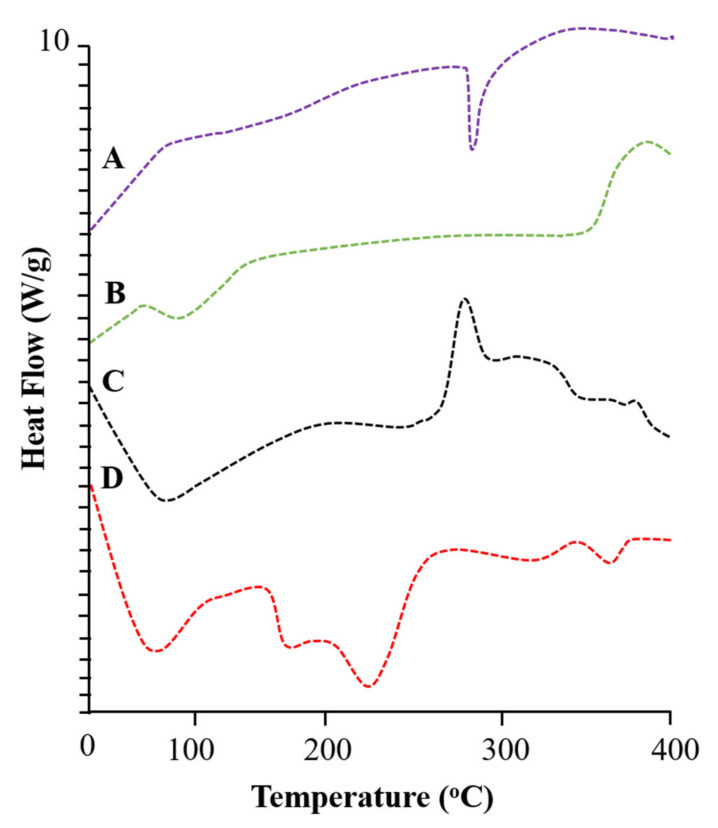
DSC thermograms of (**A**) acyclovir, (**B**) β-CD, (**C**) tamarind gum powder, and (**D**) the hydrogel formulation (TGB-12).

**Figure 4 pharmaceuticals-15-01527-f004:**
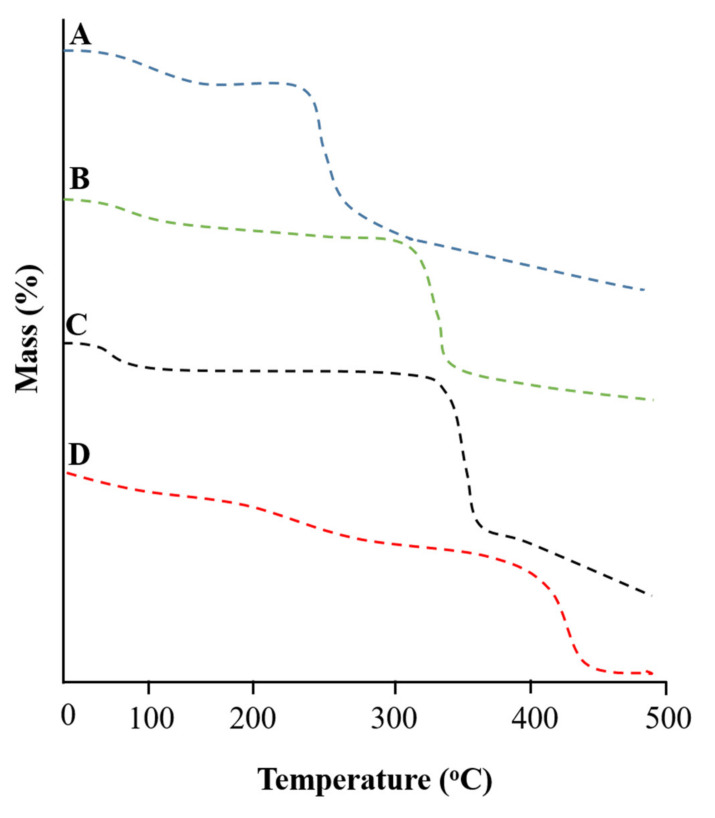
TGA thermogram of (**A**) acyclovir, (**B**) tamarind gum powder, (**C**) β-cyclodextrin, and (**D**) the developed network (TGB-12).

**Figure 5 pharmaceuticals-15-01527-f005:**
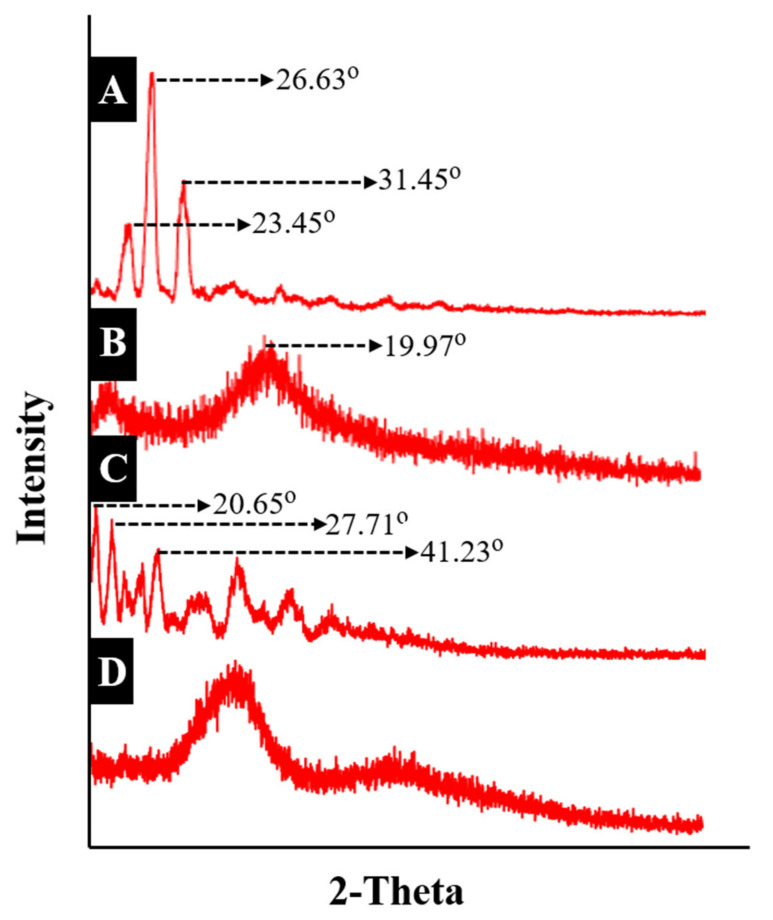
X-ray diffractogram of (**A**) acyclovir, (**B**) tamarind gum, (**C**) β-Cyclodextrin, and (**D**) the developed network (TGB12).

**Figure 6 pharmaceuticals-15-01527-f006:**
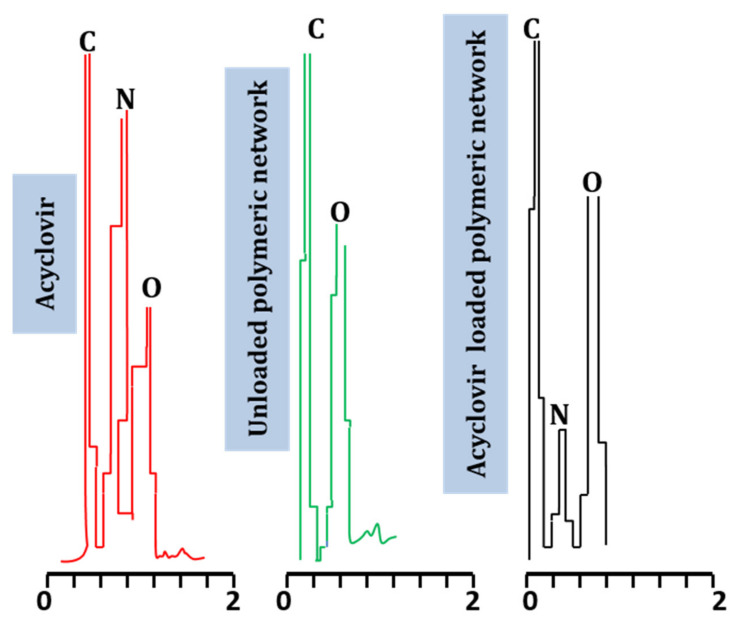
EDX spectra of acyclovir.

**Figure 7 pharmaceuticals-15-01527-f007:**
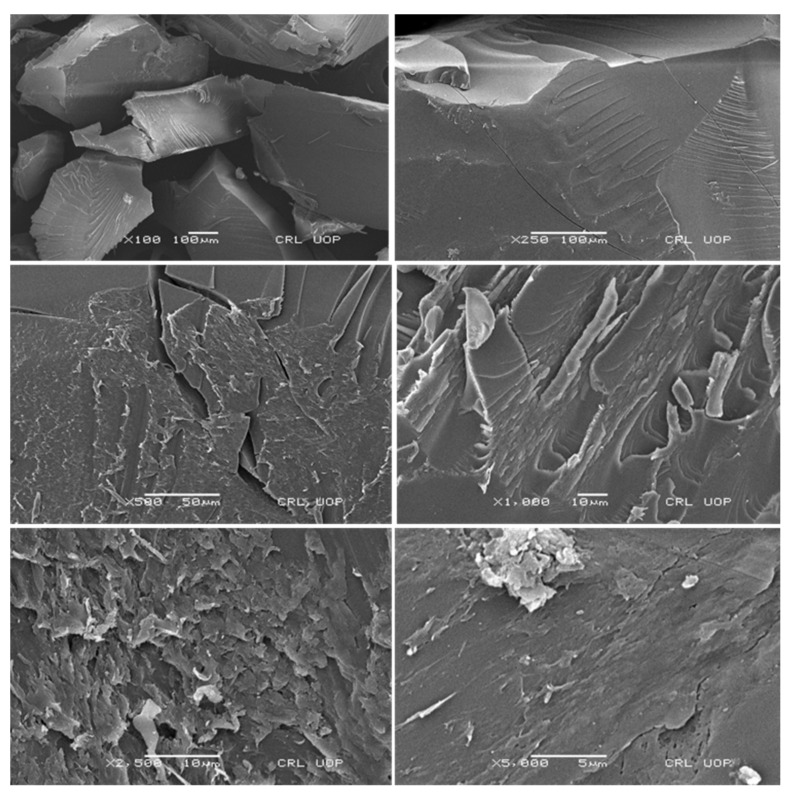
SEM photomicrographs of the developed hydrogel (TGB12) at different magnification powers.

**Figure 8 pharmaceuticals-15-01527-f008:**
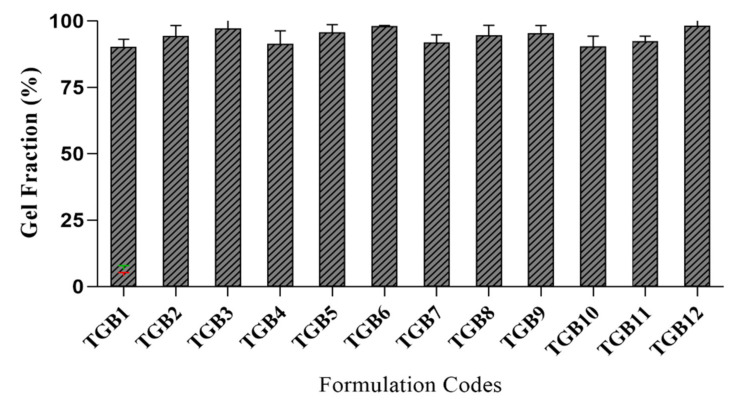
Gel fraction (%) results of hydrogel formulations (TGB1–TGB12).

**Figure 9 pharmaceuticals-15-01527-f009:**
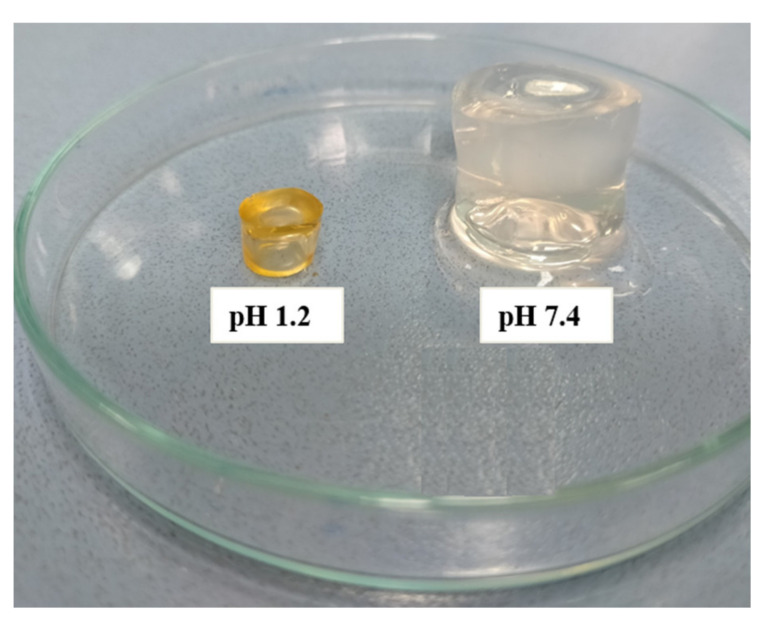
pH-responsive character of hydrogels.

**Figure 10 pharmaceuticals-15-01527-f010:**
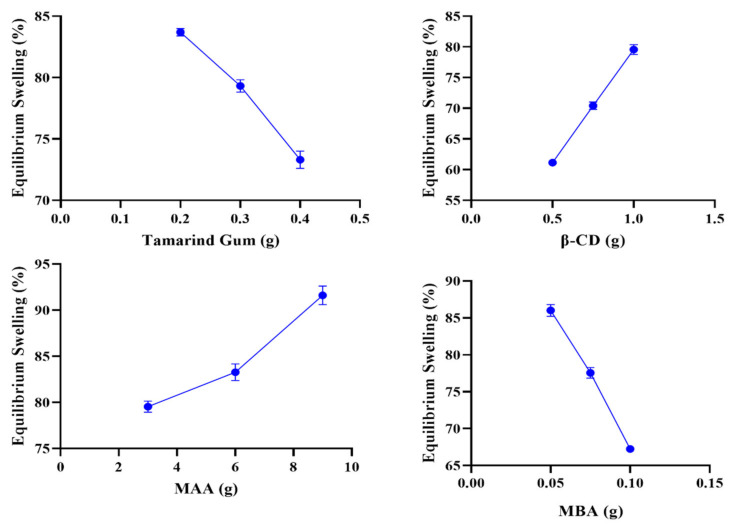
Effect of tamarind gum, β-CD, MAA, and MBA on equilibrium swelling.

**Figure 11 pharmaceuticals-15-01527-f011:**
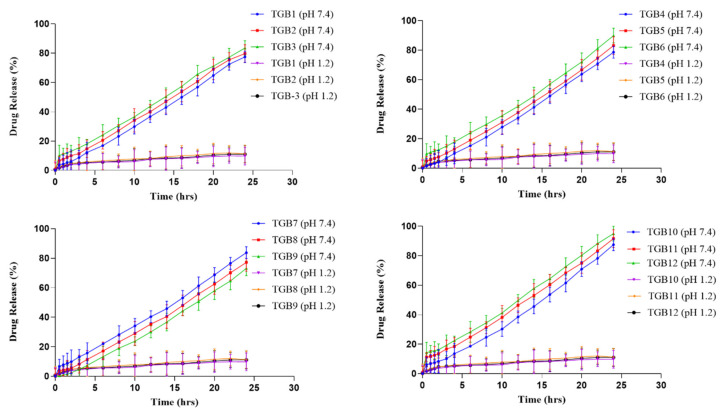
Results of ACV release from developed formulations (TGB1–TGB12).

**Figure 12 pharmaceuticals-15-01527-f012:**
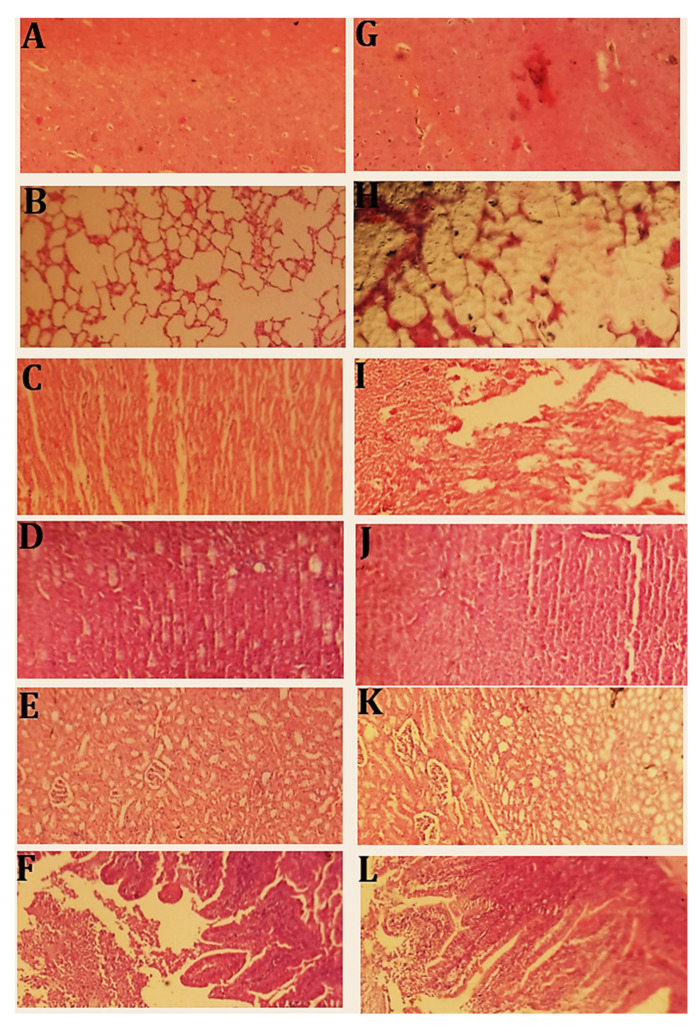
Histopathological examination of (**A**) = brain (control), (**G**) = brain (treated), (**B**) = lungs (control), (**H**) = lungs (treated), (**C**) = heart (control), (**I**) = heart (treated), (**D**) = liver (control), (**J**) = liver (treated), (**E**) = kidney (control), (**K**) = kidney (treated), (**F**) = intestine (control), (**L**) = intestine (treated).

**Figure 13 pharmaceuticals-15-01527-f013:**
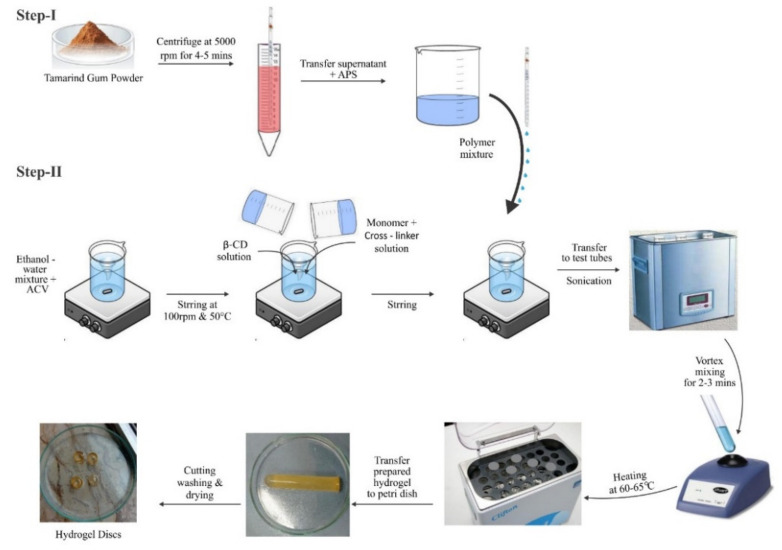
Schematic representation for the development of tamarind-β-CD-co-poly (methacrylate) hydrogels.

**Figure 14 pharmaceuticals-15-01527-f014:**
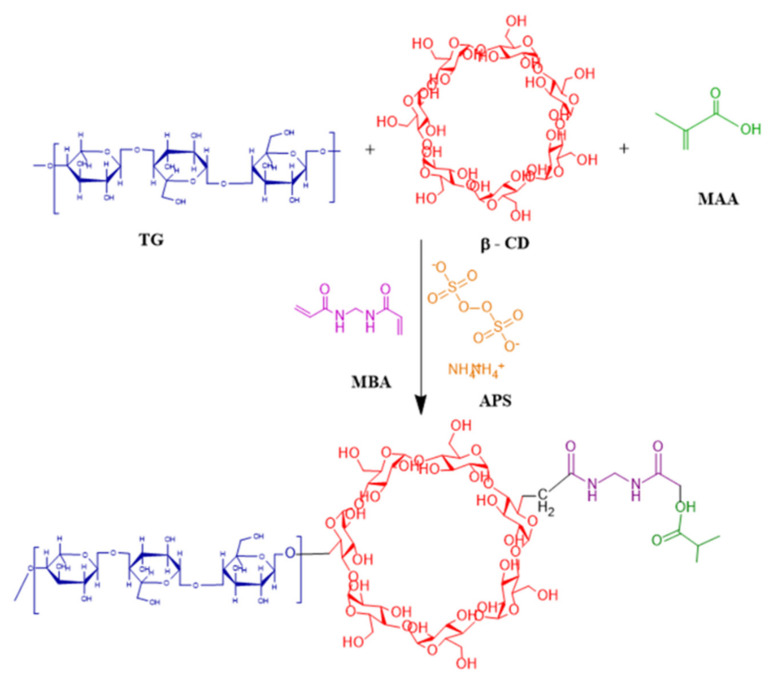
Proposed chemical structure of tamarind-β-CD-co-poly (methacrylate) hydrogels.

**Table 1 pharmaceuticals-15-01527-t001:** Elemental composition of acyclovir-loaded and -unloaded formulations.

Material Type	Element	Weight (%)	Atomic (%)
Acyclovir	C	39.85	45.39
	N	30.98	30.26
	O	28.25	24.15
Unloaded hydrogels (TGB12)	C	58.81	65.54
	O	41.19	34.46
Acyclovir-loaded hydrogels (TGB12)	C	61.16	67.80
	N	10.39	11.47
	O	38.45	32.00

**Table 2 pharmaceuticals-15-01527-t002:** Analysis of release data using kinetic models.

Kinetic Model	Parameter	Mean
Zero-order	R^2^	0.99
	K_o_	3.38
	t_25_	7.46
	t_50_	14.93
	t_75_	22.40
First-order	R^2^	0.95
	K1	0.05
	t_25_	5.86
	t_50_	14.13
	t_75_	28.27
Higuchi	R^2^	0.86
	k_H_	13.23
	t_25_	3.70
	t_50_	14.81
	t_75_	33.33
Korsemeyer–Peppas	R^2^	0.99
	K_kp_	3.56
	t_25_	7.53
	t_50_	14.81
	t_75_	22.08
	*n*	1.02

t_25_, t_50_, and t_75_ indicate the time (h) required for 25%, 50%, and 75% drug release, respectively.

**Table 3 pharmaceuticals-15-01527-t003:** Clinical observation data from oral toxicity studies.

Clinical Monitoring	Group I (Control)	Group II (Treated)
Signs of illness	None	None
**Body weight (kg)**		
1st Day	2.05 ± 0.05	2.06 ± 0.05
14th Day	2.07 ± 0.04	2.07 ± 0.04
**Food consumption (g)**		
1st Day	75.49 ± 1.52	73.93 ± 1.52
14th Day	73.66 ± 2.08	75.48 ± 3.01
**Water intake (mL)**		
1st Day	190.48 ± 3.05	195.59 ± 1.52
14th Day	204.26 ± 2.51	200.65 ± 1.15
**Dermal Toxicity**	None	None
**Ocular Toxicity**	None	None
**Mortality**	None	None

**Table 4 pharmaceuticals-15-01527-t004:** Results of hematological and biochemical analysis.

Parameter	Group I (Control)		Group II (Treated)	
	1st Day	14th Day	1st Day	14th Day
Hb (g/dL) (10–15 g/dL)	13.23 ± 0.55	13.13 ± 0.45	13.66 ± 0.50	13.33 ± 1.59
TLC (8.1–21.5 × 10^3^)	6.83 ± 0.05	6.91 ± 0.45	7.18 ± 0.28	7.12 ± 0.42
RBCs (3.8–7.9 × 10^6^/µL)	6.46 ± 0.25	6.36 ± 0.23	6.13 ± 0.35	6.25 ± 0.79
Platelets (per µL) (250–650)× 10^3^	339 ± 0.30	337 ± 0.36	325 ± 0.17	321 ± 0.05
Monocytes (0–3%)	3.46 ± 0.40	3.53 ± 0.15	3.53 ± 0.25	3.62 ± 0.17
Lymphocytes (30–70%)	65.03 ± 4.01	65.66 ± 4.16	64.66 ± 3.05	64.36 ± 2.08
MCV (50–75 fl)	63.31 ± 2.51	65.73 ± 1.58	65.31 ± 3.5	68.32 ± 1.52
MCH (18–24 pg)	22.91 ± 0.52	23.01 ± 0.62	21.86 ± 1.34	22.03 ± 0.52
MCHC (27–34 g/dL)	32.78 ± 2.74	33.04 ± 0.79	32.64 ± 0.30	31.94 ± 0.98
HCT (PCV)% (33–50)	43.11 ± 0.05	43.91 ± 0.17	49.31 ± 0.25	48.67 ± 0.36

**Table 5 pharmaceuticals-15-01527-t005:** Results of liver, kidney, and lipid profiles.

Parameter	Group I (Control)		Group II (Treated)	
	1st Day	14th Day	1st Day	14th Day
ALT (µ/L)	157.39 ± 3.41	125.48 ± 34.80	168.67 ± 14.61	102.43 ± 0.36
AST (µ/L)	72.23 ± 2.53	68.37 ± 0.51	87.36 ± 8.09	64.65 ± 1.24
Total proteins (g/dL)	5.82 ± 2.01	5.9 ± 3.08	7.13 ± 0.73	6.4 ± 3.02
Albumin (g/dL)	3.7 ± 0.05	4.2 ± 2.13	3.82 ± 0.05	4.42 ± 2.24
Globulin (g/dL)	2.4 ± 0.12	2.3 ± 0.10	2.49 ± 0.06	2.55 ± 0.08
A/G ratio (%)	1.5 ± 0.16	1.5 ± 0.51	0.83 ± 0.84	1.11 ± 0.39
Creatinine (mg/dL)	1.03 ± 0.11	0.87 ± 0.02	1.22 ± 0.44	0.98 ± 0.11
Uric acid (mg/dL)	3.27 ± 0.12	3.36 ± 0.15	2.58 ± 0.37	2.38 ± 0.53
Urea (mmol/L)	15.96 ± 3.67	19.21 ± 3.25	17.66 ± 0.52	16.98 ± 0.68
BUN (mg/dL)	16.5 ± 0.51	17 ± 0.43	12.39 ± 0.58	11.81 ± 1.10
Cholesterol (mg/dL)	36.10 ± 2.69	35.51 ± 2.89	65.33 ± 5.09	63.32 ± 4.04
Triglycerides (mg/dL)	64.59 ± 7.76	63.10 ± 0.06	67.33 ± 5.85	66.32 ± 0.21
HDL (mg/dL)	46.2 ± 0.80	47 ± 0.62	52.36 ± 1.64	54 ± 1.12
LDL (mg/dL)	64.11 ± 3.09	61.02 ± 3.01	80.27 ± 4.75	85.02 ± 4.78
VLDL (mg/dL)	14.23 ± 5.08	20.03 ± 5.04	25.34 ± 2.09	28.06 ± 2.72

**Table 6 pharmaceuticals-15-01527-t006:** Composition of tamarind/β-CD-co-poly (methacrylate) hydrogel formulations.

Formulation Code	Tamarind Gum (g)	β-Cyclodextrin (g)	N, N Methylene Bis-Acrylamide (g)	Methacrylic Acid (g)	Ammonium Persulfate (g)
TGB1	0.2	0.5	0.03	3.0	0.02
TGB2	0.3	0.5	0.03	3.0	0.02
TGB3	0.4	0.5	0.03	3.0	0.02
TGB4	0.2	0.5	0.03	3.0	0.02
TGB5	0.2	0.75	0.03	3.0	0.02
TGB6	0.2	1.0	0.03	3.0	0.02
TGB7	0.2	0.5	0.05	3.0	0.02
TGB8	0.2	0.5	0.075	3.0	0.02
TGB9	0.2	0.5	0.1	3.0	0.02
TGB10	0.2	0.5	0.03	3.0	0.02
TGB11	0.2	0.5	0.03	6.0	0.02
TGB12	0.2	0.5	0.03	9.0	0.02

## Data Availability

Data is contained within the article.

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
