# Peer review of "Development and Optimization of Tamarind Gum-β-Cyclodextrin-g-Poly(Methacrylate) pH-Responsive Hydrogels for Sustained Delivery of Acyclovir"

_pharmaceuticals, 2022, doi:10.3390/ph15121527_

Round 1

Reviewer 1 Report (New Reviewer)

The reviewed manuscript is very valuable because it comprehensively covers new acyclovir delivery system, i.e., from the development of the preparation, through the study of its physicochemical properties, to the in vivo safety assessment in an animal model. However, I have some doubts and suggestions about the manuscript that could improve the reception of the content.

- Lines 69 - 73 - the description should be presented in a more scientific way rather than in everyday language.

 The results of the FTIR, DSC and TGA studies of the pure compounds used to make the final formulation can be transferred to the supplementary information because these substances are already well known and characterized, and the additional description adds nothing new to the field.

- Please explain why the authors used TGB-12 only for FTIR, TGA, DSC, PXRD, EDX, SEM and in vivo studies.

- Line 142 and further - please clarify whether you used the IR or FTIR technique.

- Line 391 - The research cannot be classified as in vitro studies, because the presented research model does not fall within the definition of this type of research.

- Line 395 - I think the authors meant 37oC instead of 370.5oC

- Please specify what T25, T50 and T75 mean in Table 2 and how the values ​​were calculated.

- The authors should provide more information on the release data used to select a mathematical model to describe the release behavior of ACV. This is especially important to be clear for the estimation of the payload release mechanism – the authors found that the ACV release is consistent with super case II kinetics. I am not convinced that the change in the composition of the gels did not affect the release kinetics.

- Lines 437 - 439 - the payload is not "emitted" from the delivery system, and the release kinetics is not a "process".

- In the case of hematological and biochemical tests on animal blood samples, information on the reference values should also be included, as it is somewhat suspicious that the results of blood samples from rabbits treated with TGB-12 are very different from control samples from untreated animals.

Author Response

Dear Editor in Chief,

We are grateful for your valuable revisions to our paper. In the attached word document, we respond to the comments point by point. Please see the attachment.

Dr. Mounir Salem

Reviewer 2 Report (Previous Reviewer 2)

Thank you for answering the comment. If by any chance the authors don't want to separate some files to supporting information. I dont have comment for that.

Author Response

Dear Editor in Chief,

Thank you for the valuable comments. Please find the attached paper.

Dr. Mounir

This manuscript is a resubmission of an earlier submission. The following is a list of the peer review reports and author responses from that submission.

Round 1

Reviewer 1 Report

The main drawback of this manuscript relies in its formal presentation. In general, English language has to be extensively improved. All the manuscript sections, from Introduction to Results and Discussion, Materials and Methods and Conclusions need to be carefully and deeply revised and polished. Moreover, the efficacy of the drug loaded formulation have to be tested in the animal model.  In this form the work is not suitable for publication in Pharmaceuticals.  

Other points are listed below:

- Line 143-144 : It can be elaborated by a fact that all monosaccharides of tamarind have two isomers one consisting of ring structure and other comprised of linear structure. Linear structures possess glucose, xylose and galactose units. These units were liable for tretching of the carbonyl band. These sentences have to be rrephrased.

-Line 521-522: “Aqueous free radical polymerization method (Figure 13) was employed with little bit modifications for the preparation of tamarind gum based polymeric networks”. With little bit modifications in relation with…? Please rephrase.

Lines 522-525: “Extensive literature study was performed to estimate minimum and maximum quantities of each ingre-  dient to be utilized in the hydrogel formation. Moreover, different trials were performed in lab  using the quantities within the minimum and maximum quantities estimated from literature  study”. Please rephrase and cite the literature references.

-Please reformulate the conclusions section.

Reviewer 2 Report

The authors have stated that the manuscript is too described the use of hydrogel for drug delivery.

Here, we would like to give some comment to improve the work:

1. Please describe the full name of HSV before giving acronym.

2. We think that the hydrogel images in Picture is unclear. Better take the picture from the side (cross-section)

3. Please describe the position of the DSC graph for example, by giving line for the shift of the each peaks.

4. The scale bars isnt clear. 

5. Please check the writing, such as pH-responsive not pH responsive.

6. There are a lot of figure in the manuscript better separate it into main figure and supporting information.